# The Regulation of RNA Modification Systems: The Next Frontier in Epitranscriptomics?

**DOI:** 10.3390/genes12030345

**Published:** 2021-02-26

**Authors:** Matthias R. Schaefer

**Affiliations:** Centre for Anatomy & Cell Biology, Division of Cell-and Developmental Biology, Medical University of Vienna, Schwarzspanierstrasse 17, Haus C, 1st Floor, 1090 Vienna, Austria; matthias.schaefer@meduniwien.ac.at; Tel.: +43-1-40160-37702

**Keywords:** epitranscriptomics, regulation, reproducibility

## Abstract

RNA modifications, long considered to be molecular curiosities embellishing just abundant and non-coding RNAs, have now moved into the focus of both academic and applied research. Dedicated research efforts (epitranscriptomics) aim at deciphering the underlying principles by determining RNA modification landscapes and investigating the molecular mechanisms that establish, interpret and modulate the information potential of RNA beyond the combination of four canonical nucleotides. This has resulted in mapping various epitranscriptomes at high resolution and in cataloguing the effects caused by aberrant RNA modification circuitry. While the scope of the obtained insights has been complex and exciting, most of current epitranscriptomics appears to be stuck in the process of producing data, with very few efforts to disentangle cause from consequence when studying a specific RNA modification system. This article discusses various knowledge gaps in this field with the aim to raise one specific question: how are the enzymes regulated that dynamically install and modify RNA modifications? Furthermore, various technologies will be highlighted whose development and use might allow identifying specific and context-dependent regulators of epitranscriptomic mechanisms. Given the complexity of individual epitranscriptomes, determining their regulatory principles will become crucially important, especially when aiming at modifying specific aspects of an epitranscriptome both for experimental and, potentially, therapeutic purposes.

## 1. Introduction

For over half a century, chemical RNA modifications have been known to exist in RNA [1,2,3,4]. For most of this time, these modifications were considered to be chemical additions to the sequence of primarily non-coding RNAs (ncRNAs) which affected their biogenesis, stability and, likely, function. Curiously, various modifications had also been detected in coding RNAs, namely *N7*-methylguanosine (m^7^G), *N6*-methyladenosine (m^6^A), inosine (I), ribose methylation (Nm) and poly-A additions, indicating the potential for a regulatory role in messenger RNAs (mRNAs) [5,6,7,8].

Research on mRNA modifications lay largely dormant for many years with the exception of RNA editing (C-to-U and later A-to-I) due to the relatively easy detection by DNA and RNA sequencing [9,10,11,12,13,14,15,16,17,18,19]. Then, a thought-provoking commentary in 2010 boldly suggested the existence of molecular mechanisms, called “RNA epigenetics”, which, akin to epigenetics affected RNA functionally [20]. This idea was based on the earlier identification of an α-ketoglutarate-dependent dioxygenase activity (AlkB) that was able to demethylate not only m^6^A-modified DNA but also m^6^A-modified RNA [21]. This finding together with the identification of another enzymatic activity capable of removing m^6^A from RNA [22] provided the necessary spark that triggered renewed interest in chemical modifications, especially in their the regulatory potential for mRNAs. Between then and now, feverishly productive years have resulted in an impressive number of studies reporting on links between specific types of RNA modifications and many essential biological processes. Importantly, the discovery of dedicated molecular machineries that can dynamically change RNA identity led to coining the terms “epitranscriptome” and “epitranscriptomics” [23,24]. Ever since, proteins installing specific RNA modifications are called writers, activities that modify or even remove the same modification are considered to be modifiers/erasers, and proteins that interpret the RNA modification signature are called readers. Accordingly, the resulting transcriptome-wide RNA modification landscape, established by a particular epitranscriptomic machinery, is now called an epitranscriptome. There is still semantic confusion in the field. Some authors are calling only specific and reversible mRNA modifications, containing carbon-nitrogen bonds (i.e., m^6^A(m), m^1^A) as epitranscriptomic. Others use the term for any RNA modification, be it terminally or internally placed on any RNA including non-coding RNAs. Despite of these semantic issues, determining individual epitranscriptomes (through ‘epitranscriptomics’) is now a rapidly developing field that is (currently) focused on a limited number of RNA modifications (m^6^A, m^5^C, m^1^A, Ψ, m^7^G, and Nm; structural information in [25]). This limitation does exist because reliable and reproducible detection of a given RNA modification is a key requirement when studying the biological impact of RNA modifications. Hence, during the last 10 years, the major share in epitranscriptomics research was invested in the development of mass spectrometry- and sequencing-based methodologies to map specific RNA modifications [25,26,27,28]. This technological progress has resulted in extensive and transcriptome-wide studies and the publication of large data sets originating from different cell types, tissues and organisms. Importantly, RNA modifications have also been associated with human health. In particular, deregulation of particular writers, readers and modifiers/erasers, and thereby specific epitranscriptomes, has been observed in multiple human cancers and various diseases (reviewed in [29,30,31,32,33]). Rather than reiterating the extensive amount of accumulated knowledge here, the interested reader is referred to the multitude of excellent reviews written on the available technologies and the consequences of mutations in epitranscriptomic circuitry [25,26,34,35,36,37,38,39]. Instead, this article aims at highlighting some of the open questions in epitranscriptomics with a focus on the need to better understand the regulation of the enzymes that establish, modulate and erase individual epitranscriptomes. 

## 2. Key Questions in Epitranscriptomics

### 2.1. Once Is Never: How Reproducible Is Current Modification Mapping?

While the existence of RNA modifications and their effects on RNA functionality is measurable, our understanding as to how specific epitranscriptomes are established, dynamically modulated and partially erased is very much incomplete. To better define epitranscriptomic mechanisms and to be able to decipher their phenotypic consequences, one basic prerequisite is reproducibly determining individual epitranscriptomes. As of now, many of the published efforts to map specific epitranscriptomes (mapping the same RNA modification transcriptome-wide) have produced contradictory primary data sets. This has resulted in an “uneasy” debate about the limitations of current technologies and, importantly, how to draw conclusions from the existing data sets [25,40,41,42,43,44,45,46,47]. Yet constantly, an increasing number of modification mapping experiments is being published (still using debated technology [48]), along with descriptions of a staggering variety of RNA modification-dependent phenotypes. In contrast, few follow-up studies have been published that utilized the available mapping data as basis for addressing the biological impact of a specific RNA modification in a specific RNA at a specific position. Notable exceptions are studies on the functional consequences of RNA editing, which can draw on reproducible mapping data produced by multiple (and independent) laboratories [49,50,51,52]. Inaccurate mapping of any RNA modification will greatly affect all experimental conclusions, hypothesis building, and importantly, ongoing bioinformatics efforts to predict RNA modification patterns in silico [53,54,55,56,57,58,59,60,61]. A persistent question, therefore, is how to reliably map specific epitranscriptomes, not only in reproducible fashion but also sufficiently robust to technical variation.

### 2.2. Stoichiometry: How Many Molecules Need to Be Modified for Biological Impact?

The biological impact of specific RNA modifications likely depends on the percentage of individual transcripts that are modified. For instance, a modification affecting RNA stability is unlikely to have impact if only a few transcripts are modified. On the other hand, a modification such as A-to-I editing at a specific mRNA position resulting in the translation of a particular protein isoform, might have major impact even when produced at low levels. Quantitative measurements revealed that the relative levels of various eukaryotic mRNA modifications (other than RNA editing) were: 0.2 to 0.6% for m^6^A/A, 0.015 to 0.054% for m^1^A/A, 0.025 to 0.1% for m^5^C/C, 0.001–0.004% for hm^5^C/C, 0.2–0.6% for Ψ/U, and only 0.003% for m^6^Am/all nucleosides (listed and individually referenced in [27]). These are rather low values. And even though m^6^A is repeatedly mentioned as the most abundant mRNA modification, the actual values translate into 1 to 3 m^6^A-modifiations per 1000 nucleotides of an mRNA. Do these values represent nucleotides from a fraction of RNAs with the same sequence (sub-stoichiometric modification) or is every RNA molecule modified, yet is only expressed in a few cells (stoichiometric modification)? State-of-the-art liquid chromatography-mass spectrometry (LC-MS) methods allow the absolute quantification of individual RNA modifications [62,63,64,65]. However, this technology still requires relatively large (and pure) quantities of the RNA analyte, making it unsuitable for high-throughput analyses [28]. Therefore, sequencing-based methods will need to be developed for determining the stoichiometry of RNA modifications in complex and low-input samples. Methods such as site-specific cleavage and radioactive-labelling followed by ligation-assisted extraction and thin-layer chromatography (SCARLET) and m^6^A-level and isoform-characterization sequencing (m^6^A-LAIC-seq) can be used to quantify m^6^A/A levels at candidate loci and in transcriptome-wide fashion [66,67]. Furthermore, RiboMeth-Seq allows quantitative insights into ribose methylation (Nm) levels at specific sites [68]. Also, RNA bisulfite sequencing, a method allowing the mapping of m^5^C at single nucleotide resolution [69], could be used to quantify the ratio of m^5^C/C in specific transcripts. However, when using sequencing-based approaches for quantitative RNA modification measurements, it will be crucially important to include unique molecular identifiers (UMIs) during sample preparation [70], a technical detail that was often not included in existing RNA modification mapping data.

### 2.3. RNA Modification Come and Go: Constitutive or Dynamic Signatures?

Epitranscriptomics has largely been defined by the transcriptome-wide interrogation of one specific epitranscriptome at a given time. With respect to occurrence, stoichiometry and latency in the same RNA sequence, the obtained results allow conceptually separating at least two categories of epitranscriptomic signatures. In one category, RNA modifications can be detected in almost every sequence of a specific RNA class, type or species. For instance, every functional mRNA contains specific terminal RNA modifications such as m^7^G at cap structures and non-templated poly-A-tails. Individual tRNAs, rRNAs and snRNAs contain invariant RNA modifications at specific positions with high stoichiometry indicating that these modifications are constitutive (reviewed in [71,72,73,74]). In contrast, RNA modifications are also detectable at sub-stoichiometric levels suggesting that their placement is not constitutive but (dynamically) regulated. This notion is supported by observations showing that RNA modifications at specific positions can differ quantitatively, especially in response to particular cellular, developmental or environmental changes [75,76,77,78,79,80,81,82,83]. Convincing examples exist for A-to-I editing, which, while constitutive in repetitive RNAs [17,84,85], appears to be dynamically modulated in particular mRNAs [80,82,86]. Another intriguing example is the developmental stage-dependent activity of Initiator of Meiosis 4 (IME4), the yeast homolog of mammalian METTL3, mediating m^6^A only during meiosis and sporulation in *Saccharomyces cerevisiae* [87,88]. Recent evidence indicated that also microbiome-dependent regulation of eukaryotic RNA modifications. The micronutrient queuine, produced by prokaryota, is not only a precursor for substituting guanosine with queuosine (Q) in specific tRNAs [89], but placement of Q also affects m^5^C levels at specific tRNA positions [90,91]. It follows that future epitranscriptomics would gain from incorporating any context-relevant information (cellular, developmental, environmental, disease) when publishing and depositing RNA modification mapping data into databases.

### 2.4. The Multi-Substrate/Promiscuity Problem

The chemical universe of RNA modifications is rather complex but most known modification reactions involve methylation groups, followed by isomerizations and deamination reactions [92,93,94,95]. Many of the enzymes with the potential to modify RNAs have been only bioinformatically identified and await characterization [96,97,98,99]. However, various knockout or knockdown approaches addressing specific epitranscriptomic systems followed by transcriptome-wide determination of respective epitranscriptomes have revealed the substrate specificity for a select number of writers and modulators/erasers. While many enzymes modifying ncRNAs such as tRNAs, rRNAs or small nuclear RNAs (snRNAs) appear to have limited and often evolutionary conserved substrate specificity, others display pronounced substrate promiscuity. Examples for limited substrate specificity are highly conserved enzymes such as particular members of the NOP2/Sun domain (NSUN) or methyltransferase-like (METTL) family of proteins targeting single nucleotides in rRNAs [100,101,102,103,104,105], tRNAs [106,107,108] and snRNAs [109]. Another writer displaying minimal substrate specificity is Apolipoprotein B mRNA-editing enzyme 1 (APOBEC-1), which mediates the C-to-U deamination of only one position in apolipoprotein B (apoB) mRNA [110]. Expanded substrate specificity is represented highly promiscuous NSUN family members targeting various cytosolic and mitochondrial tRNAs, other small ncRNAs and hundreds of different mRNAs [111,112,113,114,115,116,117]. Similarly, various pseudo-uridine synthetases (PUS) modify miRNAs, tRNAs and mRNAs [118,119], and the METTL3/METTL14 complex addresses hundreds to thousands of different mRNAs [23,40,120], pri-miRNAs [121], long ncRNAs [122] as well as circular RNAs (circRNAs) [123]. The most extended multi-substrate RNA modification enzymes are the adenosine deaminases acting on RNA (ADARs) targeting hundreds to millions of adenosines [124]. Such substrate promiscuity complicates obtaining a more detailed understanding of a given epitranscriptomic system since it will remain unclear whether the modification of a particular position in only one RNA species or the combination of different positions in different RNAs is causative for an observed phenotype. Hence, an important but unresolved question in epitranscriptomics remains: how to experimentally address RNA modifications in specific RNAs without modulating the rest of the respective epitranscriptome?

### 2.5. Phenotypic Pleiotropy: Boon or Bust for Deciphering the Impact of Epitranscriptomes?

To gain first insights into the impact of specific epitranscriptomes on cellular processes, “early-stage” epitranscriptomics has mostly employed classic reverse genetics. Gene-specific knockout or overexpression constructs have been used to interfere with or enhance the function of various writers, readers or modifiers/erasers). Especially, one RNA modification, the addition of a methyl group onto adenosine resulting in m^6^A, has become the “flagship” modification for most of current epitranscriptomics. m^6^A writers, readers and modifiers/erasers have been genetically manipulated in different cells, tissues and organisms. The wide range of reported phenotypes indicated that modulating this particular RNA modification system is affecting literally every aspect in cell biology but raises also important conceptual questions. Just to make the point, here is a non-exhaustive list of processes affected when interfering with m^6^A RNA modification systems: proliferation [125], mRNA splicing and adipogenesis [126,127,128,129], RNA stability [76,130,131], mRNA translation and decay [75,130,132,133], mitotic entry [134], DNA damage response [135], circadian rhythm [136], signaling pathways [137], oocyte maturation [138,139], maternal-zygotic transition [140], sex determination [141,142], spermatogenesis [143,144], reprogramming to pluripotency [145,146], stem cell renewal and differentiation [147,148,149,150,151], tumorigenicity and cancers [137,149,152,153,154,155], yet also anti-tumor immunity [156], neural development and differentiation [139,157], axon regeneration [158], neurotransmitter-related circuitry [159], learning and memory [160], CNS myelination [161], neural development [157,162], miRNA processing [121] and viral infection [163,164,165,166,167]. The combination of these observations has mostly been interpreted as proof for the notion that m^6^A in RNA is necessary and crucially required for cellular functions. However, the sum of the observed phenotypes points both at experimental and conceptual problems. Specifically, the sheer scope of affected biological processes indicates that classic knockdown or knockout approaches makes it virtually impossible to separate cause from consequence. Phenotypic pleiotropy, albeit more limited, has also been observed when modulating other epitranscriptomic systems as those responsible for A-to-I editing (reviewed in [168,169]), for m^5^C (reviewed in [170]), for Ψ (reviewed in [171]) and for Nm (reviewed in [172,173]). In summary, phenotypic pleiotropy suggest that interfering with gene products with multi-substrate promiscuity and acting upstream of a complex system of effectors is not necessarily resulting in a better understanding of that particular system. How then should one interrogate the impact of an RNA modification without completely removing its writers, readers or modulators/erasers from a complex system? 

## 3. Dynamic RNA Modifications Likely Require Context-Specific Regulation

The observed substrate promiscuity of writers and modulators/erasers, the dynamic nature as well as the sub-stoichiometric levels of some RNA modifications suggest that the components of RNA modification systems are subject to regulation. This article will ignore regulatory principles such as gene expression since expression changes cannot explain how different RNA species are modified at varying stoichiometries by the same enzyme within the same cell. Hence, the following paragraphs will focus on the post-transcriptional and post-translational regulation of epitranscriptomic activities through changes in their subcellular localization, interactions with proteins and/or RNAs, protein modifications and the availability of co-factors. 

### 3.1. Subcellular Localization: Regulated or by Chance?

The enzymes establishing and modulating/erasing RNA modifications as well as their RNA substrates need to find each other within cells. While this is rather obvious, many text books still define the interior of a cell, including subcellular compartments, as spaces filled with molecules, which find each other stochastically and thereby by chance. How do epitranscriptomic systems that affect most RNAs sub-stoichiometrically come together? Most RNAs undergo either co- or post-transcriptional modification, close to the source of transcription, as part of elaborate processing and maturation steps (reviewed in [73,174]). Some RNAs, such as snRNAs and tRNAs, are not only modified in the nucleus, but become exported into the cytoplasm for additional modifications before being reimported into the nuclear compartment to undergo final processing (reviewed in [71,175,176]). These observations indicate a network of RNA modification enzymes residing in various cellular compartments. However, proteins do also dynamically change localization. Examples for intracellular trafficking of RNA modification activities are substrate-promiscuous enzymes. For instance, particular ADAR isoforms reside exclusively in the nucleus while others shuttle in and out of the nucleus or can be detected in stress- or infection-induced subcellular structures (reviewed in [177]). Another example is the complex localization pattern of NSUN2 (a promiscuous cytosine-5 RNA methyltransferase), which is mostly nucleolar but is also imported into the mitochondrial matrix [116,117]. In addition, NSUN2 re-localizes to different subcellular regions depending on the cell cycle stage and on environmental stress exposure [111,116,117,178,179]. Furthermore, individual members of the pseudo-uridine synthetase (PUS) protein family display diverse localization patterns (reviewed in [180]). And, last but not least, context-dependent and complex subcellular localization has also been revealed for writers and modulators/erasers of m^6^A (reviewed in [34]). Currently, defining the subcellular localization of the various components of epitranscriptomic circuitry is largely focused on proteins rather than RNAs. However, there is ample evidence for the regulated and dynamic localization of individual RNAs within cells (reviewed in [181]). The potential influence of RNA localization on (sub-stoichiometric) RNA modification levels is an exciting but still unexplored possibility. Hence, future attempts that aim at understanding epitranscriptomic systems might consider not only addressing the localization of writers, modulators and erasers, but also determine the localization of specific substrate RNAs. In doing so, one should preferably be implementing single molecule imaging techniques, which can distinguish single nucleotide changes/modifications such as fluorescent in-situ hybridization (FISH) techniques involving hybridization chain reaction (HCR) [182,183,184], or click-encoded rolling FISH (ClickerFISH, [185]), which could be combined with in vivo RNA localization approaches (reviewed in [186]). Combining in vivo localization of specific RNAs with the systematic analysis of spatially restricted proteomes in a particular cell type and context (reviewed in [187]) will allow determining the subcellular details of “where and when” epitranscriptomic systems act.

### 3.2. Guiding Epitranscriptomic Activities: Context-Dependent Protein-Protein Interactions

Mutating the most upstream components of various RNA modification systems (writers or modifiers/erasers) has resulted in pleiotropic phenotypes, which are hard to interpret mechanistically. In order to modulate epitranscriptomic signatures in a more targeted fashion, determining the downstream interactions of writers and modifiers/erasers would allow defining points of interference that would not necessarily result in removing an entire epitranscriptome. Besides protein-protein interactions that determine the localization of epitranscriptomic activities (and thereby their substrate choice), specific interactions could also directly inhibit or enhance of their catalytic function. Some efforts have already been made towards determining how the most abundant RNA modifications such as m^6^A, Ψ, or A-to-I could be regulated through interacting proteins. While various writers were able to modify RNAs under minimal in vitro assay conditions, not unexpectedly, those proteins acted in multiprotein complexes in vivo. For instance, the core components of the m^6^A system for modifying mRNAs (METTL3 and METTL14) form a heterodimeric writer complex, which methylate RNAs in vitro. In this complex, METTL3 is the catalytically active subunit while METTL14, unable to bind the essential co-factor S-adenosyl-methionine (SAM), plays a structural role that is critical for substrate recognition [188]. This suggests that context-dependent interactions of METTL14 could greatly affect the activity of METTL3. Importantly, interactions with the splicing factor WTAP and (so far) five more proteins (VIRMA, RBM15/RBM15B, ZC3H13 and HAKAI) are required for localization to nuclear speckles and m^6^A deposition on mRNA in vivo (reviewed in [189]). It was proposed that these accessory proteins are directing methylation specificity towards coding and non-coding RNAs through interaction with particular RNA-binding proteins [122,190]. Similar conclusions can be drawn for the A-to-I editing system, in which the function of a catalytically inactive ADAR family member (ADAR3) [191] is thought to be regulating other ADARs, for instance by binding to and blocking substrate RNAs [83]. Similarly, ADARs (A-to-I editing) and PUS (Ψ) can act as stand-alone enzymatic activities in vitro, but interact with a plethora of proteins in vivo (reviewed in [171,180,192]). While these findings are not surprising, the challenge now lies in how to disentangle this complexity in order to better understand the effects of individual protein interactors on the activity of particular writers and modifiers/erasers in vivo. In order to do so, more systematic approaches are needed, preferably by in vivo mapping using advanced proximity biotinylation and ligation techniques (reviewed in [187,193]), and in combination with cross-linking mass spectrometry, which allows delineating the interaction surfaces of interacting proteins but also reveals structural information (reviewed in [194]).

### 3.3. Non-Coding RNAs: Guides and Modulators of Epitranscriptomic Activities

RNA-guided processing or degradation of DNAs and RNAs is evolutionary conserved. For instance, bacteria and archaea produce RNAs from genomically encoded “clustered regularly interspaced short palindromic repeats” (CRISPR) [195,196], which served as guide RNAs (gRNAs) for CRISPR-associated (Cas) proteins. Cas proteins act as endonucleases, which upon being guided to complementary DNA or RNA sequences, degrade such sequences (reviewed in [197,198,199]). Particular Cas proteins can also target and degrade RNAs (reviewed in [200]). Similar to CRISPR-Cas systems, eukaryotic RNAi pathways require small RNAs (microRNAs, miRNAs; small interfering RNAs, siRNAs; piwi-associated RNAs, piRNAs) to guide particular proteins (Argonautes) towards their RNA targets resulting in sequence-specific processing or degradation of complementary nucleic acids (reviewed in [201]). 

Importantly, enzymes writing or modulating/erasing RNA modifications can interact with non-substrate RNAs resulting not only in guiding but potentially also in the modulation of their activities. A prominent example for how RNA modification enzymes are guided by RNAs was discovered in *Trypanosoma brucei* [9]. In trypanosomes, many mitochondrial RNAs undergo post-transcriptional uridine insertions and deletions as a prerequisite for the production of functional messengers. The information for these RNA editing processes is provided by trans-acting small RNAs guiding a multiprotein complex to particular positions in the mitochondrial transcriptome (reviewed in [202]). Later, it was found that gRNA-mediated RNA modifications are not restricted to protozoa nor to the process of RNA editing. A common theme for the underlying mechanisms is the assembly of multiprotein complexes that are guided by RNAs to base-pair with target RNAs and thereby direct modification of specific ribonucleotides. For instance, hundreds of gRNAs have been identified that target activities introducing Nm and Ψ into various RNAs and different species. These small gRNAs have been named small nucleolar RNAs (snoRNAs) and classified into box C/D RNAs, directing Nm and box H/ACA RNAs that target activities to introduce Ψ. A representative example is U6 snRNA, which acquires eight Nm and three Ψ for full functionality [203,204]. The introduction of such gRNA-mediated modifications are highly conserved in respect to sequence context and their existence affect the processing and function of ribosomal RNAs as well as spliceosomal snRNAs (reviewed in [205]). Importantly, small RNAs can not only guide but could also affect the activity of RNA modification enzymes on their substrate RNAs. While there is not much evidence for this scenario yet, one example is snoRNA (SNORD115), which targets Nm to an ADAR2-mediated pre-mRNA editing site, thereby specifically interfering with ADAR2 activity [206]. Furthermore, ADAR1 interacts through one of its three dsRNA binding domain with the nuclear import receptor (Transportin 1), which is mutually exclusive with binding to substrate dsRNAs [207] suggesting that the availability of specific dsRNAs can determine the localization and thereby the substrate choice of this writer. In this respect, it is also noteworthy that not all RNAs that are targeted by various writers and modifiers/erasers need to be substrate RNAs with a biological function. A case in point is the sub-stoichiometric activity of the (cytosine-5) methyltransferases NSUN2 and NSUN6 in mRNAs resembling the sequence contexts and structures of their respective tRNA substrates [114,208]. This observation raises the question as to whether these sites represent consequential mRNA modifications or are (only) off-targets, which could affect the activity of these enzymes on tRNAs. While these selected examples highlight the potential for RNAs to affect the localization, protein interactions and substrate specificity of RNA modification enzymes, the challenge lies now in complement the mapping of direct targets for a particular epitranscriptomic systems with additional RNA interactions that could have regulatory function. Approaches such as artificial intelligence-based predictions trained by chemical context profiling [209], proximity labelling in subcellular compartments combined with protein-RNA crosslinking [210,211,212,213,214] together with monitoring the activity of writers/modifiers/erasers on specific RNAs will likely uncover which RNAs affect the activity of RNA modification circuitry in a specific subcellular compartment and biological context.

### 3.4. Post-Translational Modifications of Epitranscriptomic Activities

Even though post-translational modifications (PTMs) occur in most proteins including RNA modification enzymes, specific PTMs might be an inroad into experimentally addressing the specificity and dynamic nature of epitranscriptomic activities. To date, a role of PTMs for the turnover, localization and catalytic activity of selected writers and modifiers/erasers has been reported. For instance, ADAR1 isoforms are subjected to context-dependent PTMs such ubiquitination resulting in proteasomal degradation [215], or phosphorylation facilitating exportin 5-mediated transport into the cytoplasm [216]. In addition, nuclear import of ADAR2 requires phosphorylation, which, if disturbed, results in poly-ubiquitination by E3 ligase activities and proteasomal degradation [217]. Furthermore, fat mass and obesity-associated protein (FTO), an eraser of m^6^A, can become SUMOylated at a single lysine residue, which promotes FTO degradation thereby affecting the balance between adenosine methylation and demethylation [218]. Also, direct effects on RNA modification activities have been observed. For instance, SUMOylation of ADAR1 can reduce its editing activity without causing degradation or altering the subcellular localization [219]. Furthermore, the catalytic subunit of the m^6^A methyltransferase complex (METTL3) can be modified by small ubiquitin-like modifier 1 (SUMO1) at various lysine residues both in vitro and in vivo [220]. These PTMs did neither affect METTL3 stability, localization nor the interaction with METTL14/WTAP but inhibited m^6^A deposition on mRNAs. However, the impact of many existing PTMs remains unclear. For instance, mutating individual phosphorylation sites in METTL3 or METTL14 did not affect heterodimer formation or the catalytic activity of METTL3 on model substrates [221] suggesting context-specific effects that cannot be observed in vitro. Hence, more systematic approaches need to be implemented to obtain a better understanding of the impact of PTMs on individual epitranscriptomes in vivo. This could involve combinatorial studies such as global profiling of PTMs in whole and, importantly, context-specific proteomes (reviewed in [222]). Once, context-specific PTMs on writers and modifiers/erasers are known, the generation of site-specific substitutions and structural mimics (reviewed in [223]), by using genome-editing tools, will facilitate functional in vivo studies. 

### 3.5. Co-factor Requirements and Context-Dependent Metabolic Interactions

Most known RNA modifications involve enzymatic reactions attaching specific chemical moieties to nucleic acids [95]. Importantly, these reactions require the availability of co-factors or co-substrates, many of which are provided by micronutrients (i.e., vitamins and minerals) either through dietary intake or specific microbial activities (reviewed in [224]). It follows that the availability of such co-factors and co-substrates will affect many RNA modifications. Since about 70 % of all known RNA modifications contain one or more methyl groups [93,94], methyl donors such as S-adenosylmethionine (SAM), are of utmost importance for efficient RNA methylation (reviewed in [225]). Indeed, it has been shown that limiting micronutrient levels, including SAM depletion, have major impact on cell growth [226,227,228]. In addition, SAM and its demethylation product, S-adenosyl-homocysteine (SAH), are integral molecules in the folate and methionine cycles. Products of this so-called one-carbon metabolism are crucially important for basic processes such as the biosynthesis of phospholipids, polyamines and nucleotides, amino acid homeostasis, the redox defense system, and, importantly, for nucleic acid and protein methylation (reviewed in [229]). Because of its central role in cellular functions, the one-carbon metabolism is tightly regulated. This includes feed-back and feed-forward mechanisms that respond to changes and imbalances in nutrition, stress exposure or aging [230]. Interestingly, the synthesis of SAM appears to be under the control of epitranscriptomic mechanisms. For instance, SAM depletion resulted in reduced m^6^A in the 3’ UTR of MAT2A, encoding a ubiquitous mammalian SAM synthetase, and its concomitant upregulation through splicing-dependent mRNA stabilization [231,232]. Furthermore, manipulation of the cytosine-5 RNA methyltransferase NSUN2 resulted in changes in the methionine and tricarboxylic acid (TCA) cycles as well as synthetic pathways for amino acids [233]. Specifically, higher levels of methionine and SAM were observed in *NSUN2* mutant cells indicating changes in the output of the methionine cycle [233]. In addition, RNAs can interact with various metabolic enzymes (reviewed in [234]). The reason for these interactions are not completely understood but it has been proposed that RNAs (and their modification status) could act as sensors for cellular changes that require metabolic adjustments (reviewed in [92,235]). These observations underscore the intriguing complexity involving RNAs, epitranscriptomic activities and metabolic pathways (reviewed in [236]). Some even might explain the phenotypic pleiotropy when manipulating epitranscriptomic systems since metabolite-mediated (secondary) effects including epigenome and gene expression changes have been reported [237]. To start disentangling the (direct or indirect) interplay of RNA modification systems with metabolic pathways, more sophisticated experimentation than gene knockout studies followed by gene expression analyses will be required. This might include context-dependent manipulation of particular writers or modifiers/erasers, for instance by targeted protein degradation (reviewed in [238]), and the concurrent measurement of metabolic and gene expression patterns by combining single cell transcriptomics and metabolomics (reviewed in [239]).

## 4. The Next Frontier: Getting a Closer Look at the Details

The last 10 years have defined yet another frontier in molecular biology, the existence of epitranscriptomes. Early efforts in epitranscriptomics have been awarded with regular publicity, mostly for developing technologies that allow mapping individual epitranscriptomes at transcriptome-wide fashion. In addition, first insights into the impact of specific epitranscriptomic systems have produced a breath-taking picture of their immense complexity. However, both the accumulation of epitranscriptomic mapping data as well as the multitude of phenotypes resulting from malfunctioning RNA modification systems, have caused another kind of scientific competition. Rather than paving the way for more detailed and in-depth studies of specific (and known) RNA modification signatures, the amount of data and its complexity appears to deter from being used when formulating and testing new hypotheses. This is at odds with the immense amount of compiled mapping data and some of the precision tools now available that would allow “digging into the data”. Rather than doing that, a large fraction of the field focuses on only one particular RNA modification system, the m^6^A system modulating mRNAs, and reports with regularity on its involvement in anything that resonates with the notion that biology does not work without m^6^A-modified mRNAs. Another fraction is in “discovery mode” continuously “hunting” for “novel” RNA modifications by mass spectrometry or sequencing-based approaches. In contrast, a minority fraction in the field is conducting experiments aimed at investigating the impact of particular RNA modifications in specific RNAs. Some examples for the latter are studies on the impact of A-to-I editing at specific sites in specific RNAs in particular human disease models [240,241,242], the analysis of how specific RNA modifications affect innate immune responses (reviewed in [243]), or the impact of chemical modifications on RNA structures [244,245,246,247]. These approaches are facilitated by the recent development of in vivo methodology allowing site-specific introduction of RNA modifications [248,249,250,251,252,253] or their removal [251,254,255,256], by the use of nucleoside analogues (reviewed in [257]) and through variations of single molecule imaging techniques allowing to query positional information on modified RNAs [258]. A guiding example as to how continuous investment in deciphering the molecular details of the impact of particular RNA modifications can result in amazing knowledge leaps is the recent breakthrough for RNA-based therapeutics including the development and approval of mRNA vaccines, which could not have happened without focusing on the impact of particular modifications on RNA stability and interactions with the innate immune system [259,260,261,262,263,264,265,266].

Is the use of modified mRNAs in vaccine development all of what RNA modifications can teach us about biology, the ingenuity of human adaptability, technological progress and disease? Likely not. However, in order to move epitranscriptomics from an emerging and experimentally tractable phenomenon to one that can be better understood, the experimental focus needs to divert from counting numbers to addressing the mechanistic details of the modification reactions preferably with atomic scale resolution. To do so, better definitions of the biological, developmental and environmental context in which particular epitranscriptomic systems modulate RNA identity need to be incorporated into experimental designs. Furthermore, the context-dependent regulation of specific RNA modification activities needs to be systematically addressed, preferably by utilizing the great variety of established vertebrate and non-vertebrate model organisms, which offer many advantages over mammalian (cancer) cells constantly (evolving) in culture. Most importantly, structural knowledge will be required as the very prerequisite for an in-depth understanding of any RNA modification system. While some studies have addressed the structural basis for the activity of particular RNA modification enzymes, largely focusing on m^6^A and Ψ circuitry, more efforts will have to be made to determine the structures of other RNA modification enzymes, preferably in combination with their respective RNA substrates. This will result in arriving at a more solid understanding of the structure-function relationships between RNAs and enzymes that determine the complexities of individual epitranscriptomes.

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
