# Peer review of "The Regulation of RNA Modification Systems: The Next Frontier in Epitranscriptomics?"

_genes, 2021, doi:10.3390/genes12030345_

Round 1

Reviewer 1 Report

I do have somesuggestions that may improve the manuscript even further.

  1. To me, the title is somewhat misleading, as it made me believe to be reading a perspective on how noncoding RNAs (the regulators) are regulated themselves by their modifications. The title did not make me think of the modifiers, which is the main topic of the paper. I would adjust the title in order to attract the right audience.
  2. Although the author aims to focus on RNA-modifiers, a simple overview of the most common RNA modifications, and their function, would be helpful for the reader. This overview, wether graphic, or in table-form, could then be combined be an overview of known writers, erasers and readers of these different modifications.
  3. The use of English is excellent. That said, I am an advocate of the use of simple straightforward sentences, to ensure that the scientific message comes across to all readers, including non-native English speakers. Here and there, single sentences cover 6-8 lines of text. I feel the readability of the paper could be greatly enhanced by shortening some of these very longf and complex sentences.
  4. Section 2.1 focuses mostly on the reliability of data generated on modifications themselves, whereas the rest of section 2 focuses on the modifiers. This seems inconsistent to me.
  5. The introduction to section 3 is somewhat confusing and not entirely to the point. I think in general, section 3 would be helped greatly by a graphic illustration of intracellular localisations.
  6. In the introduction of Section 3, the author claims that the modifier are mostly stationary, but later contradicts this. The contradiction makes more sense to me than the initial claim.
  7. For me, section 3.3 is a little too far off-topic. All protein are subject to PTMs, this is not specific to epitranscriptome writers and erasers. Instead of this section, I feel the author could include a section on RNA modifiers, like snoRNAs. Although they are mentioned here and there, many modifcations depend on guide RNAs to guide writer proteins to their target sites. This aspect of epitranscriptome biology is completely ignored in this perspective.
  8. Althoughh SNORD115 was initially found in brain tissue, this snoRNA is by no means brain-specific (Jorjani et al, Nucleic Acids Rearch, 2016).
  9. In section 4, I appreciate that the author takes a stand and aims to point future research in the direction of more detailed functional studies. I do feel that maybe the expressed opinion on existing research is a little bit too negative. Perhaps that tone could be made a bit more sympathetic.
  10. The author claims that more detailed studies of RNA modifications and their fuction in relevant bioglogical models should be perfomred, but does not cite any such studies, even though they are being performed, including the work by Stellos et al (Nat Med, 2016), Jain et al (EMBO J, 2018 and van der Kwast et al (Circ Res 2018, Mol Ther 2019 and Mol Ther Nucleic Acids 2020) on A to I editing, isomiR formation and 2'Ome. 

Author Response

please, see the attachment

Reviewer 2 Report

There are quite a lot of excellent reviews about epitranscriptomics these days. In this Perspective, the author chose a unique aspect of this important field by focusing on the regulatory mechanisms of RNA modification enzymes. The key questions in epitranscriptomics are well-thought out. Overall, the Perspective was well-written and prepared in an informative manner. 

My main concern for the current version is that the author did not mention at all about m6A editing approach, which has been reported quite broadly by many groups. This is relevant to the author's statement that "how to experimentally address the biological impact of RNA modifications in specific RNAs without modulating the rest of the respective epitranscriptome." 
